# Comparison of Azathioprine-Induced Pancreatitis and Gastrointestinal Intolerance in IBD: Role of Demographics, Clinical Variables, and HLA DQA1/DRB1 Alleles

**DOI:** 10.3390/jcm14238539

**Published:** 2025-12-02

**Authors:** Tugce Eskazan, Oguz Kagan Bakkaloglu, Murat Toruner, Haluk Tarik Kani, Bilger Cavus, Volkan Yilmaz, Nalan Gulsen Unal, Ozlen Atug, Burhan Cagcag, Mehtap Dogruel, Erkan Yilmaz, Filiz Akyuz, Yusuf Ziya Erzin, Ali Ibrahim Hatemi, Aykut Ferhat Celik

**Affiliations:** 1Division of Gastroenterology, Department of Internal Medicine, Cerrahpasa Faculty of Medicine, Istanbul University-Cerrahpasa, İstanbul 34098, Turkey; o.k.bakkaloglu@gmail.com (O.K.B.); dryusuferzin@yahoo.com (Y.Z.E.); ihatemi@yahoo.com (A.I.H.); aykutferhatcelik@gmail.com (A.F.C.); 2Division of Gastroenterology, Department of Internal Medicine, Ankara University Faculty of Medicine, Ankara 06620, Turkey; murattoruner@yahoo.com (M.T.); yilmazvolkan@ankara.edu.tr (V.Y.); 3Division of Gastroenterology, Department of Internal Medicine, Marmara University Faculty of Medicine, İstanbul 34722, Turkey; drhtkani@googlemail.com (H.T.K.); ozlen.atug@marmara.edu.tr (O.A.); 4Division of Gastroenterology, Department of Internal Medicine, İstanbul University Faculty of Medicine, İstanbul 34093, Turkey; dr_bilgercavus@yahoo.com (B.C.); filizakyuz@hotmail.com (F.A.); 5Division of Gastroenterology, Department of Internal Medicine, Ege University Faculty of Medicine, İzmir 35040, Turkey; drnalanunal@gmail.com; 6HLA Tissue Typing Laboratory, Cerrahpasa Faculty of Medicine, Istanbul University-Cerrahpasa, İstanbul 34098, Turkey; burhan.cagcag@iuc.edu.tr (B.C.); mehtap.dogruel@iuc.edu.tr (M.D.); erkanyilmaz@klu.edu.tr (E.Y.)

**Keywords:** azathioprine, inflammatory bowel disease, acute pancreatitis, gastrointestinal intolerance, HLA-DQA1, HLA-DRB1, pharmacogenetics

## Abstract

**Background:** Azathioprine (AZA)-associated acute pancreatitis (AP) and gastrointestinal intolerance (GI-INT) are major causes of drug discontinuation in inflammatory bowel disease (IBD). This study compared HLA alleles, demographics, and clinical variables between AZA-AP and AZA-GI-INT. **Methods:** Data from five IBD centers included control (*n* = 88), AZA-AP (*n* = 44), and GI-INT (*n* = 44) groups. AP was defined by the Atlanta criteria, and GI-INT as acute dyspeptic symptoms related to AZA that resolved after withdrawal. Demographics, disease features, and HLA-DQA1/DRB1 alleles were assessed for associations. **Results:** Among 176 patients, female sex was more frequent in AZA-AP and GI-INT than controls (*p* = 0.018, *p* < 0.001). AZA-AP patients were older at diagnosis vs. controls (*p* = 0.016) but not vs. GI-INT (*p* = 0.15). Smoking and alcohol were more common in AZA-AP. The median onset of AP was four weeks, with 91% occurring within three months. GI-INT occurred rapidly, with a median of one day and a maximum of three days after the first dose. HLA-DQA1/DRB1 positivity was comparable in GI-INT and controls (9.2% vs. 14.8%, *p* = 0.42) but higher in AZA-AP (27.3% vs. 14.8%, *p* = 0.08). Regression identified female sex, smoking, alcohol, budesonide, and HLA-DQA1/DRB1 positivity (OR 3.01, 95% CI 1.004–9.058; *p* = 0.049) as independent risk factors for AZA-AP. **Conclusions:** AZA-AP, but not GI-INT, appears genetically influenced, with HLA-DQA1/DRB1 association extending across populations. In IBD, AZA-AP usually emerges within three months and is linked to female sex, smoking, alcohol, and budesonide. GI-INT typically develops within hours to three days of initiation. These findings support AZA-AP and GI-INT as distinct idiosyncratic entities shaped by genetic, metabolic, and sensitivity factors.

## 1. Introduction

Inflammatory bowel diseases (IBD) are chronic immune-mediated disorders of the gastrointestinal tract. Prevalence is highest in North America, Australia, and Scandinavia, with a marked north–south decline in Europe, reaching lower rates in Italy, Greece, and Türkiye [1].

Immunomodulators, particularly azathioprine (AZA), remain a key component in long-term IBD management, even in the biologic era. However, AZA-related adverse events (AZA-ADRs) cause discontinuation in up to 26% of patients [2]. While dose-dependent events such as cytopenia or mild liver enzyme elevations may respond to dose adjustment, others, like acute pancreatitis (AP) and gastrointestinal intolerance (GI-INT) are dose-independent and, to the best of our knowledge, classified as idiosyncratic adverse effects, and require drug withdrawal [3].

The incidence of AP is higher in IBD than in the general population, often linked to IBD therapies, particularly AZA and, less commonly, 5-aminosalicylic acid (5-ASA). Reported rates of AZA-AP range from 1.9 to 6.1% in retrospective studies [4,5], up to 7.3% in prospective cohorts [3], with a pooled prevalence of 3.8% in a systematic review [6]. In a Japanese case–control study, 1.9% of 366 IBD patients developed AZA-AP, while none occurred in 298 AZA-treated non-IBD controls [5]. Similarly, a Dutch study reported AZA-AP in 1.5% of autoimmune hepatitis and 0.4–0.5% of post-transplant patients, lower than the 4.5% seen in an IBD cohort [7]. These conditions may have an inherently higher predisposition to AP than IBD itself.

AZA-AP usually develops within 2–4 weeks of AZA initiation in IBD, typically within the first two months at any given dose [2,8,9]. The course is generally mild and self-limiting, resolving rapidly after drug withdrawal [3]. Because of recurrence risk, AZA is not reintroduced, and IBD management usually proceeds with biologics.

On the other hand, GI-INT is a more frequent AZA side effect (6–13%) [4,10] and is not limited to IBD [11]. It usually arises within hours to days of the first dose, presenting with abdominal pain, nausea, and sometimes mild vomiting. These early-onset symptoms should be clinically differentiated from acute pancreatitis (AP), where laboratory testing or imaging is reserved for atypical or delayed presentations.

In our prior study on AZA-AP, budesonide use and disease activity (particularly in Crohn’s disease) showed some association, but active smoking was the only independent risk factor, conferring a nearly three-fold increased risk [12]. The interplay between these factors and Crohn’s disease (CD) itself, however, poses a cause–effect dilemma and complicates interpretation.

Nevertheless, the pathogenesis of AZA-AP remains unclear. Its timing and presentation suggest a T-cell-mediated delayed-type hypersensitivity reaction [13]. Pharmacogenetic studies show a strong link with HLA-DQA102:01 and HLA-DRB107:01 [14], supporting the potential of pre-treatment genetic screening to identify high-risk patients and reduce adverse events. It remains unclear whether these HLA alleles are universally applicable as a pre-assessment tool for AZA-AP. The primary aim of this study was to reassess the predictive value of HLA-DQA1 and HLA-DRB1 haplotypes as a clinical screening tool for AZA-AP in IBD, considering also AZA-GI-INT and non-HLA factors. A secondary aim was to determine whether AZA-GI-INT shares the same haplotypes with AZA-AP or represents a distinct entity.

## 2. Methodology

### 2.1. Study Design and Patient Selection

This multicentre retrospective study included data from five tertiary centers in Türkiye. The study population consisted of patients diagnosed and treated in tertiary IBD centers. These institutions represent different geographic regions of the country (northwest, central, and western Türkiye), covering a large population base and thus improving the representativeness of the study cohort.

Patient enrolment was completed in May 2025. Since this was a retrospective study, the medical records of all patients who had received azathioprine for IBD were comprehensively reviewed across the five participating centers to identify eligible cases.

IBD patients meeting inclusion criteria were enrolled. Those who developed AZA-AP or AZA-GI-INT were contacted, and consenting participants provided peripheral blood samples during outpatient visits for further analysis.

### 2.2. Inclusion Criteria

#### 2.2.1. AZA-AP and AZA-GI-INT Groups

Patients were eligible for the AZA-AP and AZA-GI-INT groups if they had the following: (i) a confirmed CD or ulcerative colitis (UC) diagnosis by endoscopy, histopathology, and radiology; (ii) AZA use for maintenance or induction; and (iii) development of AP or GI-INT during AZA treatment. AP was defined per the revised Atlanta criteria [15] as persistent severe abdominal pain during AZA therapy with amylase/lipase ≥ 3× upper limit and/or imaging findings consistent with AP.

#### 2.2.2. Gastrointestinal Intolerance

As literature definitions were lacking and mostly emphasized acute/severe features, GI-INT to AZA was defined as follows: patient refusal to continue AZA due to acute severe dyspeptic symptoms (nausea, abdominal pain or distension, and varying degrees of general discomfort) with rapid resolution after discontinuation.

#### 2.2.3. Control Group

Controls were IBD patients with confirmed diagnosis by endoscopy, histopathology, and radiology, treated with AZA ≥ 6 months for induction or maintenance without adverse events. The cohort was divided in a 2/1/1 ratio: Control/AZA-AP/AZA-GI-INT.

We also retrospectively reviewed HLA typing from 97 healthy kidney donors at Cerrahpasa Faculty of Medicine to assess background HLA-DRB1 allele frequency. This provided context for genetic predisposition in patients versus a healthy population. As per routine pre-transplant protocol, only HLA-DRB1 subtyping was available.

### 2.3. Data Collection and Analysis

For all patients, recorded data included the following: age, sex, diagnosis, disease location, duration, height, weight, smoking, alcohol, comorbidities, and concomitant drugs. Additional AP-related factors (gallstones, hyperlipidemia) were retrospectively reviewed. AZA duration, initial dose, and dose at AP or GI-INT onset were also documented. Active smoking was defined as regular cigarette consumption (≥1 cigarette/day). For patients in the AZA-AP group, smoking status was recorded at the time when AP developed. For patients with AZA-GI-INT, smoking was assessed at the time of GI-INT onset. In the control group, active smoking referred to patients who continued to smoke during the follow-up period under AZA treatment.

AP severity, hospitalization, biochemical markers, and imaging findings were assessed. As controls lacked an event for reference, comparison with the AZA-AP group was limited to treatments within 12 weeks of AZA initiation, about three times the median time to AZA-AP onset.

The impact of demographics, disease features, and HLA-DQA1/HLA-DRB1 status on AZA-AP and GI-INT was analyzed. Variables with *p* < 0.1 for AP were further tested in multivariate regression.

### 2.4. Blood Collection, DNA Extraction, and HLA Typing Methods

Whole blood samples (10 mL) were collected from patients and controls. DNA was extracted from 3 mL EDTA blood samples using the EZ1 DNA extraction kit with the Bio Robot EZ1 system (Qiagen, Hilden, Germany) and stored at −70 °C until analysis.

Low-resolution (2-digit) HLA-DR and HLA-DQ typing was performed with LIFECODES SSO kits (Immucor, Dreieich, Germany) on a Luminex 100/200 instrument. The method is based on a reverse sequence-specific oligonucleotide assay using probes on fluorescent microspheres.

PCR was carried out in 50 µL reactions containing Lifecodes Master Mix, genomic DNA, and Taq polymerase, followed by hybridization with probe mixes and detection with streptavidin–phycoerythrin on the Luminex 200 system (Luminex Corp., Austin, TX, USA). Results were compared with the IMGT/HLA database (v3.11.0) using MatchIT DNA software (Immucor, Norcross, GA, USA, v3.56). 

### 2.5. Statistical Analysis

Statistical analyses were performed with SPSS v25.0 (IBM, Armonk, NY, USA). Categorical variables were given as frequencies/percentages, and continuous as mean ± SD or median (min–max). Categorical comparisons used Chi-square or Fisher’s exact test, continuous variables used Student’s *t*-test or Mann–Whitney U test as appropriate. Univariate analyses identified potential risk factors for AP, with variables at *p* < 0.1 entered into logistic regression. Associations between HLA alleles and AP were expressed as odds ratios (ORs), comparing AP with both GI-INT and controls. A *p* value < 0.05 was considered statistically significant in all analyses.

## 3. Results

A total of 176 patients were included in the study. Patients were divided into the following three groups in a 2:1:1 ratio: Control group (*n* = 88), AZA-AP group (*n* = 44), and AZA-GI-INT group (*n* = 44).

### 3.1. Patient Characteristics

Female sex was more frequent among patients who developed AZA-related adverse events. When compared to the control group (35.2%), females were more prone to develop both AZA-AP (56.8%) and AZA-GI-INT (65.2%) (*p* = 0.018 and *p* < 0.001, respectively). Crohn’s disease (CD) was the predominant diagnosis in all groups, though its frequency was slightly lower in AZA-AP (68.2%) compared to AZA-GI-INT (81.8%) and controls (80.7%), without reaching statistical significance.

The median age at diagnosis was higher in AZA-AP [39 years (IQR 17.2)] than in controls [31 years (IQR 16.5), *p* = 0.016], but not different from AZA-GI-INT [34.5 years (IQR 14.7), *p* = 0.15].

The mean initial azathioprine dose ranged between 1.5 and 2.5 mg/kg across all groups, with mean values of 58.2 mg/day in AZA-AP and 57.3 mg/day in AZA-GI-INT, showing no significant difference. In the AZA-AP group, the mean dose at onset of pancreatitis was 95 ± 35.99 mg/day.

These socio-demographic and treatment characteristics, together with HLA-DQA1/DRB1 allele status, are detailed in Table 1.

### 3.2. Time of Onset of Adverse Events

In our clinical practice, AZA is initiated at 50 mg/day and titrated to the maximum over 6–8 weeks. In our study, among patients in the AZA-AP group, the time from azathioprine initiation to the onset of acute pancreatitis ranged from 2 to 424 weeks, with a mean duration of 4 weeks, and 91% developed symptoms within 3 months.

Two late cases developed AP at eight and five years post-initiation; lacking other risk factors or dose changes, both were classified as AZA-AP and included.

The onset of AZA-GI-INT group was observed within 1–3 days of AZA initiation (mean ± SD: 1.2 ± 1 days). Most occurred within hours of the first dose, though exact timing was not consistently available, so 24 h was set as the minimum onset for analysis. As most patients discontinued AZA by phone or without physician approval, they were symptom-free at the first scheduled visit. Thus, detailed symptom history was the primary basis for AZA-GI-INT assessment.

At the time of GI events (AP, GI-INT), no patients in any group, including controls, had biliary colic, gallstones, peptic ulcer, or proton pump inhibitor/histamine-2 receptor antagonist (PPI/H2RA) use. No AP or GI-INT occurred in any group after AZA cessation during long-term follow-up.

Additional data related to AZA treatment characteristics, including duration of therapy and AZA dose at event onset across study groups, are presented in Appendix A.

### 3.3. Clinical Findings Related to Adverse Effects

Per Atlanta criteria, all AP cases were classified as mild edematous; none developed necrosis or organ failure. Two patients required hospitalization (mean stay 5 days, range 3–10); none needed intensive care. The main AZA-GI-INT symptom was acute severe dyspepsia, almost always with various levels nausea, resolving in all patients after AZA withdrawal.

### 3.4. Associated Factors

Regarding risk factors, 38.6% of patients in the AZA-AP group were active smokers. Smoking exposure was observed in 50%, 59%, and 49.4% of the AZA-AP, GI-INT, and control groups (*p* = 0.95 vs. 0.29), respectively. There was no difference in budesonide use between the AZA-GI-INT and control groups (9.3% vs. 12.5%, respectively, *p* = 0.59), while the AZA-AP group had significantly higher budesonide use (40%, *p* < 0.001). Treatment-related parameters and their comparative analyses are presented in Table 1.

### 3.5. HLA DQA1/DRB1 Allele Results

Genotypic analysis focused on HLA-DQA102:01 and HLA-DRB107:01. DQA1*02:01 positivity was 18.2% in AZA-AP, 9.2% in AZA-GI-INT, and 14.8% in controls (*p* = 0.61 and *p* = 0.36, respectively).

DRB1*07:01 positivity was 9.1% in AZA-AP and 4.5% in controls but absent in AZA-GI-INT. This allele was also not detected in 97 healthy kidney donors at our transplant center.

HLA-DQA1/DRB1 positivity was similar between the AZA-GI-INT and control groups (9.2% vs. 14.8%, *p* = 0.42) but tended to be higher in the AZA-AP group (27.3% vs. 14.8%, *p* = 0.08) (Table 1).

### 3.6. Treatment and Outcomes

No AP patients were re-exposed to AZA or other thiopurines; all were switched to biologics (anti-tumor necrosis factor (TNF) or vedolizumab). Logistic regression showed female sex, smoking, alcohol, budesonide, and HLA-DQA1/DRB1 positivity as independent risk factors for AZA-AP (Figure 1). HLA risk haplotype carriers had ~three-fold higher risk (OR 2.7, 95% CI 1.004–9.058; *p* = 0.049).

## 4. Discussion

In this study, we conducted a comprehensive evaluation of two distinct adverse events—AZA-AP and AZA-GI-INT—that developed during AZA treatment in patients with IBD.

Our findings show distinct differences between AZA-AP and AZA-GI-INT in clinical features and risk factors. The HLA-DQA102:01–HLA-DRB107:01 haplotype was a strong risk factor for AZA-AP but not for AZA-GI-INT. HLA-DQA1/DRB1 allele frequency was higher in AZA-AP (27.3%) than in GI-INT (14.8%) and controls (9.2%). These alleles were significant, particularly between AZA-AP and controls in regression analysis. The relatively higher HLA-DQA1/DRB1 positivity in controls (IBD without AZA-ADRs) versus kidney donors may partly explain why only IBD patients are vulnerable to AZA-AP [16].

Genetic predisposition to AZA-AP was previously reported in a 2014 multicentre Genome-Wide Association Study (GWAS), which demonstrated that the HLA-DQA102:01–HLA-DRB107:01 haplotype significantly increases the risk of AZA-AP [17]. The study estimated that avoiding AZA in homozygous carriers would require genotyping 76 patients to prevent 1 AP case [17].

These findings were validated in a UK/Canada retrospective cohort of 13 AZA-AP cases and 360 controls [14]. AP risk was highest in homozygous C/C versus heterozygous (A/C) or wild type (A/A). In a subsequent prospective screening study at the same centers, avoiding AZA in carriers of the HLA-DQA1–HLA-DRB1*07:01 A>C variant reduced AZA-AP prevalence from 3.4% to 0.3% [14].

While pre-treatment HLA-DQA1–HLA-DRB1 screening shows promise for preventing AZA-AP, feasibility is uncertain due to population allele frequency differences and lack of consensus on clinical use. A Swedish study reported that HLA-DQA102:01 and HLA-DRB107:01 were significantly associated with AZA-AP in IBD versus population controls (OR 3.97, 95% CI 1.57–9.97; *p* = 0.0035). Interestingly, the association was significant only in CD, with no clear link in UC [16]. This suggests that HLA-related risk may be stronger in CD, though the lack in UC could reflect small sample size. Our earlier study on non-genetic risk factors also showed CD predominance in AZA-AP (66% vs. 44% UC) [12]. In this current study, CD comprised ~70% of AZA-AP and 80% of AZA-GI-INT. This may reflect higher AZA use in CD or center-specific CD/UC ratios, yet the CD predominance may indirectly support their vulnerability to AZA-AP or GI-INT. This CD dominancy aligns with Swedish data, further suggesting that HLA-DRB1*07:01 is linked to CD-related AP [16]. In our study, the frequencies of CD were comparable across groups; therefore, diagnosis was not included in the multi-variant analysis, which could have further clarified the potential association. Since AZA-related GI intolerance has also been reported in patients with rheumatoid arthritis [11], this suggests that AZA-GI intolerance is not disease-specific.

Smoking, an environmental risk factor for both AZA-related and unrelated AP in IBD, is also a well-known CD risk factor [3,12,18] and may partly explain CD accumulation. No HLA data exist for AZA-GI-INT, yet its lowest allele positivity—absent in kidney donors (HLA-DRB1*07:01)—suggests that, like AZA-AP, both AZA-GI-INT and even controls (IBD without AZA-ADRs) may share IBD-related HLA backgrounds.

Our study is notable in showing a similar HLA association in the Turkish population. The HLA-DQA102:01–DRB107:01 haplotype represents a distinct MHC class II pattern with a certain population frequency. In Europe, DRB1*07:01 occurs in ~10–15% [19]. While HLA polymorphism–AZA adverse event links, particularly AP, are well documented in European cohorts, including Swedish and multinational GWAS [14,16], such data have been lacking in Türkiye. Although not comparable to population-based GWAS, none of the 97 healthy kidney donors carried this allele. Whether donors represent a distinct subgroup or not, IBD prevalence is higher in Nordic countries than in southern Europe and Türkiye, possibly reflecting underlying genetic differences, including HLA distributions. Our findings suggest that AZA-AP is driven by similar alleles as in high-prevalence countries, though with different rates. This implies that combined environmental and genetic burdens contribute to AZA-AP. Detection of the same alleles in our Turkish cohort indicates they are more universal, not strictly geographic or ethnic, even if overlapping with IBD-related backgrounds.

AZA-AP likely represents an immune-mediated reaction, with HLA class II molecules central to CD4^+^ T-cell antigen presentation [20]. Recent advances in immunogenetics have clarified HLA-associated idiosyncratic drug reactions [21]. It is hypothesized that AZA or its metabolites may bind within the peptide-binding cleft of class II HLA molecules, modifying the self-peptide landscape presented to CD4^+^ T cells [14]. The resemblance of AZA-AP to autoimmune pancreatitis supports this mechanism, which, if validated, may broaden understanding of HLA-associated drug hypersensitivity [16,22,23].

The severity of symptoms in AZA-GI-INT significantly affects quality of life and almost always lead to immediate AZA discontinuation [4,24]. Data remain limited due to the lack of a standardized definition, and no reliable clinical or genetic predictors exist. GI-INT is typically defined as severe dyspeptic symptoms linked to AZA after excluding common diagnoses with dyspeptic futures, like AP, acute gallbladder/biliary-related conditions, or peptic ulcer. Although not all our patients were fully evaluated for these possibilities, the very acute onset and rapid resolution of symptoms without recurrence reassured us to consider these cases as AZA-related GI intolerance.

Although GI-INT may be considered clinically more benign than AP, it makes a significant impact on quality of life; therefore, managing AZA-related GI intolerance can be very challenging in clinical practice. In a prospective study, severe nausea was the leading cause of AZA discontinuation, accounting for 8% [2]. Some patients may tolerate AZA with dose splitting or brief interruption, but others require alternative immunosuppressants. In our study, none opted to continue AZA.

The mechanism of AZA-induced nausea, vomiting, and abdominal discomfort remains unknown, and no clinical or genetic predictors exist. In our study, the lack of increased HLA risk allele frequency in GI-INT patients suggests a distinct, yet unidentified, pathophysiological mechanism [25].

Some AZA-intolerant patients may tolerate 6-mercaptopurine (6-MP) or benefit from allopurinol co-therapy, which shifts metabolites and enables lower AZA dosing [2,4,24]. In one study of 149 patients, 57% with GI intolerance tolerated 6-MP [26]. These findings support the hypothesis that pre–6-MP metabolites may drive AZA-related GI intolerance.

Regression analysis in our previous study identified female sex, older age at diagnosis, smoking, and alcohol as independent AZA-AP risk factors [12]. Female sex and smoking are already established risks [3,18]. These findings emphasize the need to distinguish smoking’s effect on IBD course from its direct role in AZA-AP development.

Because of inconsistent reports, smoking’s effect on IBD course is unclear, though it increases CD (particularly ileal) and relapse while decreasing UC risk. The elevated AP risk in AZA-treated smokers more likely reflects direct pancreatic toxicity. Smoking alters pancreatic secretions, impairs microcirculation, and increases oxidative stress, potentially amplifying drug-induced subclinical inflammation. The association between CD and smoking has long been noted in AZA-AP [3,12]. Smoking stimulates a Th1 cytokine profile, dominant in CD [27], as shown in animal studies [28], a proposed mechanism for CD worsening [29]. Thus, smoking may aggravate both CD course and AZA-AP risk via overlapping mechanisms. In smoking IBD patients, particularly CD, considered for AZA, close monitoring for AZA-AP is needed, and smoking cessation should be strongly advised.

As mentioned above, our previous study showed that active smoking is an independent AZA-AP risk factor, with ~three-fold higher risk in IBD (OR 3.208, 95% CI 1.192–8.632; *p* = 0.0035). This was quite in line with the findings of this current study, except concomitant budesonide use, which was non-significant in our prior study in multivariate analysis but remained significant here [12].

In this current study, budesonide and/or anti-TNF use was higher in AZA-AP than in controls. After multivariate analysis, only budesonide remained significant (OR 4.03, 95% CI 1.126–12.87; *p* = 0.019), while anti-TNF did not (*p* = 0.094). A previous study from Germany also identified concomitant budesonide as an independent risk factor [3]. Although they could not explain this finding, they emphasized the issue.

In our study, a group of patients continued taking budesonide and anti-TNF even after cessation of AZA; however, no additional pancreatitis occurred thereafter in either the AZA-AP or control groups. This may also support the idea that acute AZA-AP is only a product of AZA use. However, budesonide, as an independent risk factor, may still facilitate it. Although no established metabolic interaction between budesonide and AZA has been reported, AZA may have an interaction through cytochrome P450 (CYP3A4), which is also known to metabolize budesonide [30]. We may speculate that their concurrent use may influence hepatic enzymatic activity or drug clearance, potentially leading to increased accumulation of AZA metabolites. This hypothetical mechanism could contribute to AZA-AP.

Unlike AZA, used in both CD and UC, budesonide in Türkiye is available only as 3 mg capsules, mainly for distal ileal/ileocecal CD. Thus, its accumulation in CD cases may reflect treatment preference. If CD is linked to AZA-AP through genetic (HLA or non-HLA) or environmental (smoking) factors, our budesonide association may represent an overlapping reflection of CD’s HLA connections, which themselves could predispose to AP with or without AZA. We cannot further elaborate on budesonide but advise caution regarding its AP potential in IBD. Pure IBD-related AP is rare, and identifying its treatment-free association is difficult. Triggers like AZA or budesonide, along with genetic or environmental factors, may be required to provoke clinically manifest AP in IBD patients. In our study, controls had a much longer AZA follow-up (Table 1, ~70 weeks vs. 4-week median AZA-AP onset), reassuring us they were not vulnerable to AZA-AP.

A comprehensive review published in 2022 noted that, in addition to AZA, 5-ASA may also rarely cause AP. The review also highlighted that the risk of AP may be slightly increased in IBD patients due to the disease itself, depending on the population studied [31]. In our study, 5-ASA use rates were similar in AP and GI-INT groups, and no AP recurred after AZA withdrawal, supporting that AP was specific to AZA exposure.

Pharmacogenetic markers like thiopurine S-methyltransferase (TPMT) and Nudix Hydrolase 15 (NUDT15) predict myelosuppression but are not clearly linked to AZA-AP or intolerance, which are dose-independent, with distinct mechanisms [32,33]. Evidence on TPMT and AZA-ADRs is inconsistent.

This first comparative study of AZA-AP and AZA-GI-INT confirms that they are distinct entities with different frequencies, presentations, and risk factors. AZA-AP, though less frequent, is clinically more serious and strongly linked to genetic susceptibility, particularly HLA-DQA102:01–DRB107:01. Our findings in a Turkish cohort parallel the literature, suggesting that these HLA-driven risks are universal rather than geography-specific. Because of overlapping alleles with IBD’s genetic background, frequencies may be higher in IBD-prevalent regions. Environmental cofactors like smoking and CD subtype may add further risk. Budesonide’s role in AZA-AP warrants clarification in future studies. By contrast, AZA-GI-INT is more common but appears HLA-independent. Both AZA-AP and GI-INT are not AZA dose-dependent and more likely reflect idiosyncratic reactions shaped by metabolic variability and drug sensitivity.

## 5. Strengths and Limitations

Strengths of our study include a substantial multicentre cohort with comparative analysis of AZA-AP, AZA-GI-INT, and controls in IBD. It also represents the first data from the Europe–Asia bridging zone in a Turkish population. Integration of HLA genotyping with clinical factors (smoking, budesonide, CD) and discussion of AZA-AP and GI-INT pathogenesis under current literature enhances the translational relevance of our findings.

Nonetheless, some limitations should be noted:

First, the retrospective design may carry bias and missing data. Second, focusing on HLA alleles may overlook other genetic contributors, though our aim was to find out a simple HLA link rather than complex variants. Third, control group genotyping was limited; ideally, larger AZA-tolerant cohorts would better validate associations. However, our findings are still aligned with the literature.

Lastly, as our cohort included only Turkish patients, applicability to other populations may be limited. Still, the findings concur with European studies, suggesting that the HLA link in AZA-AP extends beyond geographic or ethnic boundaries.

Conclusions: Our results support that AZA-AP, but not AZA-GI-INT, may be genetically driven under environmental triggers, with the HLA link extending beyond geographic or ethnic boundaries. As distinct entities, both are AZA dose-independent and likely reflect idiosyncratic reactions shaped by metabolic variability and drug sensitivity.

Likewise, beyond clarifying budesonide’s role as an independent AZA-AP risk factor, future studies should assess how to integrate these findings into routine clinical decision-making and evaluate the cost-effectiveness of HLA screening.

## Figures and Tables

**Figure 1 jcm-14-08539-f001:**
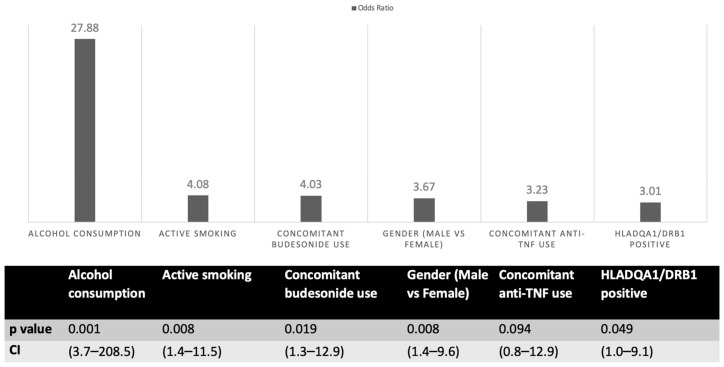
Regression analysis of factors associated with azathioprine-induced acute pancreatitis. TNF: tumor necrosis factor.

**Table 1 jcm-14-08539-t001:** Demographic characteristics, disease findings, and HLA allele status of the groups.

	Control *n*: 88 (%)	AZA-AP *n*: 44 (%)	GI-INT *n*: 44 (%)	Control vs. AZA-AP *p* Value	Control vs. GI-INT *p* Value
**Gender (Male)**	64.8	43.2	34.1	**0.018**	**<0.001**
**Diagnosis (CD)**	80.7	68.2	81.8	0.11	0.87
**Disease Involvement**					
Ileocolonic	62	66.7	47.2		
Ileal	22.5	26.7	36.1	0.71	0.63
Colonic	9.9	6.7	11.1		
**Disease Involvement–UC**					
Pancolitis	64.7	71.4	25	0.65	0.13
Left-sided colitis	29.4	28.6	62.5		
**Age at diagnosis (median-IQR)**	31(16.5)	39 (17.2)	34.5 (14.7)	**0.016**	0.15
**Smoking**					
Active use	21.6	38.6	27.3	**0.04**	0.47
Exposure (active or past smoker)	49.4	50	59.1	0.95	0.29
**Alcohol use**	2.3	13.6	9.1	**0.017**	0.1
**PSC**	1.1	2.3	2.3	1	1
**Family history of IBD**	6.8	11.4	6.8	0.37	1
**Age at AZA initiation (median-IQR)**	35 (15)	39.5 (16.75)	36 (15)	0.20	0.42
**Time from diagnosis to AZA initiation, week**	10 (36)	0 (14.7)	10 (34)	0.42	0.85
**Initial AZA dose, mean ± SD (mg/kg)**	0.95 (0.8)	1.1 (1)	0.8 (0.3)	0.20	0.19
**Concomitant medications during AZA therapy**					
5-ASA	90.9	95	95.3	0.42	0.37
Anti-TNF	6.8	19.5	34.9	**0.031**	**<0.001**
Budesonide	12.5	40	9.3	**<0.001**	0.59
**HLA allele status**					
DQA1 positivity	14.8	18.2	9.2	0.61	0.36
DRB1 positivity	4.5	9.1	0	0.3	0.3
DQA1/DRB1 (any positivity)	14.8	27.3	9.2	**0.08**	0.42

CD: Crohn’s disease; UC: ulcerative colitis; PSC: primary sclerosing cholangitis; IBD: inflammatory bowel disease; AZA: azathioprine; 5-ASA: 5-aminosalicylic acid; TNF: tumor necrosis factor; SD: standard deviation; IQR: interquartile range. bold formatting is used to indicate statistically significant parameters (*p* < 0.05).

## Data Availability

No new data were generated or analyzed. The current data underlying this article will be shared upon reasonable request to the corresponding author.

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
