# Peer review of "Comparison of Azathioprine-Induced Pancreatitis and Gastrointestinal Intolerance in IBD: Role of Demographics, Clinical Variables, and HLA DQA1/DRB1 Alleles"

_jcm, 2025, doi:10.3390/jcm14238539_

Round 1
Reviewer 1 Report
Comments and Suggestions for Authors
This article is relatively reasonable and reliable. The aim of this study was to investigate the link between Azathioprine -associated acute pancreatitis and gastrointestinal intolerance in inflammatory bowel disease by analyzing post clinical data. It is a topic of interest to the researchers in the related areas, but the article had problems.
- Which country and region are the participating patients from, and is the sample representative?
- The number of samples were sufficient to represent clinical significant?
- The relevant experimental methods and PCR reagents are not mentioned in the manuscript, and no attachments have been found. Please provide the supplementary information.
- Based on which indicators is the screening of smoking patients confirmed?
Author Response
Dear Editors and Reviewers,
We sincerely thank you for your thoughtful and constructive feedback on our manuscript entitled “Comparison of Azathioprine-Induced Pancreatitis and Gastrointestinal Intolerance in IBD: Role of Demographics, Clinical Variables, and HLA DQA1/DRB1 Alleles.’’ We appreciate the time and expertise you have devoted to reviewing our work.
We have carefully considered all comments and revised the manuscript accordingly. Below, we provide a point-by-point response to each suggestion, outlining the changes made and clarifying relevant aspects of our study.
Reviewer 1
This article is relatively reasonable and reliable. The aim of this study was to investigate the link between Azathioprine -associated acute pancreatitis and gastrointestinal intolerance in inflammatory bowel disease by analyzing post clinical data. It is a topic of interest to the researchers in the related areas, but the article had problems.
1.Which country and region are the participating patients from, and is the sample representative?
Response 1: We thank the reviewer for this valuable comment. The study population consisted of patients diagnosed and treated in five tertiary inflammatory bowel disease centers in Türkiye. These institutions represent different geographic regions of the country (northwest, central, and western Türkiye), covering a large population base and thus improving representativeness. This information has been added to the Methods (Study Design and Patient Selection) section.
‘’This multicentre retrospective study included data from five tertiary centres in Türkiye. The study population consisted of patients diagnosed and treated in tertiary IBD centers. These institutions represent different geographic regions of the country (northwest, central, and western Türkiye), covering a large population base and thus improving representativeness of the study cohort.’’
2.The number of samples were sufficient to represent clinical significant?
Response 2: We appreciate the reviewer’s question regarding sample size.
During the study period, all patients who met the inclusion criteria and developed azathioprine-induced acute pancreatitis (AZA-AP) across the five participating centers were included in the study. After identifying these cases, patients with azathioprine-related gastrointestinal intolerance (AZA-GI-INT) were selected in a 1:1 ratio, and azathioprine-tolerant controls were selected in a 2:1 ratio. Consequently, the total study cohort comprised 176 patients (44 AZA-AP, 44 GI-INT, and 88 controls).
This design ensured inclusion of all AZA-AP cases meeting the inclusion criteria, thereby maximizing representativeness despite the inherent rarity of this adverse event. While we acknowledge that larger prospective studies could further increase statistical power, our multicentre inclusion and full-case approach provide clinically meaningful data consistent with previous studies of similar or smaller size.
3.The relevant experimental methods and PCR reagents are not mentioned in the manuscript, and no attachments have been found. Please provide the supplementary information.
Response 3: We thank the reviewer for this valuable comment. The details regarding blood collection, DNA extraction, and HLA typing have now been added to the main text under the Methods (Data Collection and Analysis) section to improve transparency and readability. Specifically, we included the description of the EZ1 DNA extraction kit (Qiagen, Germany), the LIFECODES SSO typing kit (Immucor, Germany), and the PCR and hybridization procedures performed on the Luminex 100/200 system.
4.Based on which indicators is the screening of smoking patients confirmed?
Response 4: We appreciate this valuable comment. In the revised version, we clarified how smoking status was defined and assessed for each study group.
Active smoking was defined as regular cigarette consumption (≥1 cigarette/day). For patients in the AZA-AP group, smoking status was recorded at the time when acute pancreatitis developed. For patients with AZA-GI-INT, smoking was assessed at the time of gastrointestinal intolerance onset. In the control group, active smoking referred to patients who continued to smoke (≥1 cigarette/day) during the follow-up period under azathioprine treatment.
This information has been added to the ‘’Data Collection and Analysis’’ section.

Reviewer 2 Report
Comments and Suggestions for Authors
The study examines whether genetic, demographic, and clinical factors influence the development of azathioprine-associated acute pancreatitis and gastrointestinal intolerance in patients with inflammatory bowel disease.
The topic is relevant. The findings have clinical implications for optimizing azathioprine use in IBD as personalized therapy.
Several limitations should be acknowledged: The sample size is relatively small (n=44 per adverse event group), which limits the statistical power of the findings. Only HLA-DQA1/DRB1 alleles were analyzed, restricting the scope of the genetic evaluation.
My recommendations:
1. The Methods section lacks key details about the patient characteristics (e.g., age distribution, gender ratio) and the study period. Please include a more comprehensive description of the patient population and study timeframe.
2. Tables and Figures
Table 1 is difficult to interpret and should be reorganized for greater clarity and readability. The manuscript currently includes no figures. Visual presentation would significantly enhance understanding of the results.
Table 2 could be converted into a figure or diagram -for example, a pie chart showing percentage distributions, which would better illustrate the comparative results.
In the Associated Factors and HLA DQA1/DRB1 Allele Results sections, the inclusion of graphs or bar charts would make the data presentation clearer and more engaging.
3. The text would benefit from clearer numbering of main points and subpoints for better organization.
4. There is no abbreviation list, despite the frequent use of abbreviations throughout the text. Please add a List of Abbreviations section.
5.The references are not formatted according to the journal’s required citation style and should be corrected.
In my opinion: Overall
This is a clinically significant study that provides valuable insight into the distinct mechanisms underlying azathioprine -associated acute pancreatitis and gastrointestinal intolerance. However, to strengthen the manuscript, the authors should:
Expand and clarify the methodology,
Reorganize and format table 1,
Add figures to illustrate key findings,
Introduce consistent numbering and an abbreviation section, and
Revise the reference list to match the journal’s format.
Author Response
Dear Editors and Reviewers,
We sincerely thank you for your thoughtful and constructive feedback on our manuscript entitled “Comparison of Azathioprine-Induced Pancreatitis and Gastrointestinal Intolerance in IBD: Role of Demographics, Clinical Variables, and HLA DQA1/DRB1 Alleles.’’ We appreciate the time and expertise you have devoted to reviewing our work.
We have carefully considered all comments and revised the manuscript accordingly. Below, we provide a point-by-point response to each suggestion, outlining the changes made and clarifying relevant aspects of our study.
Reviewer 2
The study examines whether genetic, demographic, and clinical factors influence the development of azathioprine-associated acute pancreatitis and gastrointestinal intolerance in patients with inflammatory bowel disease.
The topic is relevant. The findings have clinical implications for optimizing azathioprine use in IBD as personalized therapy.
Several limitations should be acknowledged: The sample size is relatively small (n=44 per adverse event group), which limits the statistical power of the findings. Only HLA-DQA1/DRB1 alleles were analyzed, restricting the scope of the genetic evaluation.
My recommendations:
The Methods section lacks key details about the patient characteristics (e.g., age distribution, gender ratio) and the study period. Please include a more comprehensive description of the patient population and study timeframe.
Response: We appreciate this observation. In the revised version, we have expanded the Methods (Study Design and Patient Selection) section to include additional information on patient demographics (age distribution, gender ratio), study timeframe, and baseline clinical features. Specifically, we clarified that patient enrollment was completed in May 2025, and since this was a retrospective study, the medical records of all patients who had received azathioprine for inflammatory bowel disease were comprehensively reviewed across five tertiary IBD centers in Türkiye to identify eligible cases.
‘’Female sex was more frequent among patients who developed AZA-related adverse events. When compared to the control group (35.2%), females were more prone to develop both AZA-AP (56.8%) and AZA-GI-INT (65.2%) (p = 0.018 and p < 0.001, respectively). Crohn’s disease (CD) was the predominant diagnosis in all groups, though its frequency was slightly lower in AZA-AP (68.2%) compared to AZA-GI-INT (81.8%) and controls (80.7%), without reaching statistical significance.
The median age at diagnosis was higher in AZA-AP [39 years (IQR 17.2)] than in controls [31 years (IQR 16.5), p = 0.016], but not different from AZA-GI-INT [34.5 years (IQR 14.7), p = 0.15]. The mean initial azathioprine dose ranged between 1.5–2.5 mg/kg across all groups, with mean values of 58.2 mg/day in AZA-AP and 57.3 mg/day in AZA-GI-INT, showing no significant difference. In the AZA-AP group, the mean dose at onset of pancreatitis was 95 ± 35.99 mg/day.’’
These socio-demographic and treatment characteristics, together with HLA-DQA1/DRB1 allele status, are detailed in Table 1. All these data have now been explicitly described in the Methods section to enhance methodological transparency and reproducibility.
- Tables and Figures
Table 1 is difficult to interpret and should be reorganized for greater clarity and readability. The manuscript currently includes no figures. Visual presentation would significantly enhance understanding of the results.
Response:We thank the reviewer for this helpful suggestion. Table 1 has been reorganized for improved readability. We simplified the layout, grouped related variables together, and adjusted spacing and alignment to enhance clarity. To make the results more visually accessible, we added a new figure.
Table 2 could be converted into a figure or diagram -for example, a pie chart showing percentage distributions, which would better illustrate the comparative results. In the Associated Factors and HLA DQA1/DRB1 Allele Results sections, the inclusion of graphs or bar charts would make the data presentation clearer and more engaging.
Table 2 was converted into a figure to better illustrate the percentage distribution of the variables.
- The text would benefit from clearer numbering of main points and subpoints for better organization.
Response: The entire manuscript has been restructured with numbered main sections and subheadings to enhance logical flow and reader navigation.
- There is no abbreviation list, despite the frequent use of abbreviations throughout the text. Please add a List of Abbreviations section.
Response: Thank you for your suggestion. A paragraph listing and explaining all abbreviations used in the study has been added after the Discussion section to improve readability and clarity.
5.The references are not formatted according to the journal’s required citation style and should be corrected.
Response: We appreciate this important comment. The entire reference list has been reformatted according to the Journal of Clinical Medicine (MDPI) guidelines.
In my opinion: Overall
This is a clinically significant study that provides valuable insight into the distinct mechanisms underlying azathioprine -associated acute pancreatitis and gastrointestinal intolerance. However, to strengthen the manuscript, the authors should:
Expand and clarify the methodology,
Reorganize and format table 1,
Add figures to illustrate key findings,
Introduce consistent numbering and an abbreviation section, and
Revise the reference list to match the journal’s format.
Response: We sincerely thank the reviewer for their constructive feedback, which greatly improved the quality and readability of our manuscript. All suggested changes have been implemented, and the revised version reflects these improvements.

Round 2
Reviewer 2 Report
Comments and Suggestions for Authors
I am satisfied with the changes.